# Immunohistochemistry-based investigation of MYC, BCL2, and Ki-67 protein expression and their clinical impact in diffuse large B-cell lymphoma in upper Northern Thailand

**Phuttirak Yimpak**[1], **Kanokkan Bumroongkit**[1], **Adisak Tantiworawit**[2], **Thanawat Rattanathammethee**[2], **Sirinda Aungsuchawan**[1], **Teerada Daroontum**[3]*

1 Department of Anatomy, Faculty of Medicine, Chiang Mai University, Chiang Mai, Thailand, 2 Department of Internal Medicine, Division of Hematology, Faculty of Medicine, Chiang Mai University, Chiang Mai, Thailand, 3 Department of Pathology, Faculty of Medicine, Chiang Mai University, Chiang Mai, Thailand

* mewteerada@gmail.com

**Data Availability Statement:** All relevant data are within the manuscript.

## Abstract

Diffuse large B-cell lymphoma (DLBCL) is an aggressive type of non-Hodgkin lymphoma (NHL) that accounts for approximately 25–40% of all NHL cases. The objective of this study was to investigate the protein expression, clinical impact, and prognostic role of MYC, BCL2, and Ki-67 in Thai DLBCL patients. A retrospective analysis was conducted on 100 DLBCL patients diagnosed between January 2018 and December 2019. Immunohistochemistry was used to assess the expression of MYC, BCL2, and Ki-67. The study revealed a significant association between extranodal involvement and positive cases of MYC and BCL2. MYC expressions were associated with Ki-67 expression, while BCL2 positivity was associated with the non-germinal center B-cell (non-GCB) subtype. However, there were no significant differences in the three-year overall survival (OS) and three-year progression-free survival (PFS) rates when using cut-off points of $\geq$ 40% for MYC, $\geq$ 50% for BCL2, and $\geq$ 70% for Ki-67. Notably, DLBCL cases with co-expression of MYC and BCL2 exhibited significantly inferior three-year OS compared to other cases (0% vs. 53%; p = 0.020). Multivariate analysis identified age $\geq$ 60 years and Eastern Cooperative Oncology Group (ECOG) performance status as independent prognostic factors. In conclusion, MYC, BCL2, and Ki-67 expression can serve as prognostic biomarkers; however, their prognostic value may vary based on the specific cut-off values used. Therefore, determining the appropriate threshold for each biomarker based on individual laboratory analyses and clinical outcomes is crucial.

## Introduction

Diffuse large B-cell lymphoma (DLBCL) is the most common type of adult non-Hodgkin's lymphoma (NHL) which is represented by the heterogeneous entity based on its biological characteristics and clinical outcomes. The incidence of DLBCL is approximately 25–40% of all

**Funding:** This work was supported by the Faculty of Medicine, Chiang Mai University, grant number ANA-2563-07765, No. 160/2564. The funders had no role in study design, data collection, and analysis, decision to publish, or preparation of the manuscript.

**Competing interests:** The authors have declared that no competing interests exist.

NHL cases [1]. The international prognostic index (IPI) is an effective clinical tool to risk stratify DLBCL patients at the time of first diagnosis using five clinical parameters (Age, LDH level, Ann Arbor stage, extranodal site, and ECOG performance status). However, the biological spectrum of DLBCL is not highlighted. Gene expression profiling studies have provided not only a major division based on the cell of origin of DLBCL, whether it is a germinal center B-cell (GCB) like type or an activated B-cell (ABC) like type, but also prognostically significant information [2]. However, in Thailand, routine implementation of molecular classification is not widely available in most hospitals. As a result, alternative algorithms using immunohistochemistry (IHC) have been considered. Han's algorithm is utilized to classify DLBCL into GCB and non-GCB, based on the immunohistochemical staining of three protein biomarkers: CD10, MUM1, and BCL6. Although these two subtypes can predict the prognostic difference in DLBCL patients, they still exhibit heterogeneity and encompass various patient subgroups [3].

In 2016, the World Health Organization (WHO) classification underwent a revision, identified GCB and ABC as unique molecular subtypes of DLBCL. Additionally, a new entity called high-grade B-cell lymphoma (HGBL) was introduced. This entity is characterized by the presence of *MYC* and *BCL2* and/or *BCL6* gene rearrangements and is referred to as HGBL-double-hit (DH)/-triple-hit (TH) and commonly referred to as double-hit (DHL) or triple-hit lymphoma (THL) [4]. In addition, there is the term "dual or double expressor" lymphoma (DEL) that refers to MYC/BCL2 or MYC/BCL6 protein co-expression and triple expressor lymphoma (TEL) that refers to MYC/BCL2/BCL6 protein co-expression by IHC. A diagnosis of DHL and THL is only determined following the results of a molecular cytogenetics test called fluorescence in situ hybridization (FISH) technique. However, the WHO emphasizes that there is no consensus on which large B-cell lymphomas should undergo the FISH technique [4]. Evaluating FISH in all cases clearly may increase diagnostic costs. Therefore, to eliminate that issue, some strategies suggest limiting FISH testing by collecting cases that are positive for MYC expression and/or high Ki-67 expression by using IHC as a screening procedure [5]. Previous studies demonstrated that there is no significant difference in overall survival (OS) between DHL and DEL, indicating that DHL and DEL share similar biological behaviors [6, 7]. In 2022, the 5th edition of the WHO Classification of Haematolymphoid Tumours (WHO-HAEM5) was released, representing a molecular approach in DLBCL classification, retaining the GCB/ABC distinction and the use of IHC due to its routine application. Next-generation sequencing has revealed a complex genetic landscape in DLBCL, not otherwise specified (NOS), with numerous recurrent mutations but no unifying genetic framework, making it premature to incorporate these findings into the new classification. The term high-grade B-cell lymphoma with dual rearrangements of MYC and BCL2 and/or BCL6 from the previous edition of WHO has been conceptually reframed and reassigned to diffuse large B-cell lymphoma/high-grade B-cell lymphoma with MYC and BCL2 rearrangements (DLBCL/HGBL-MYC/BCL2). Lymphoid neoplasms with dual MYC and BCL6 rearrangements have been excluded from the DLBCL/HGBL-MYC/BCL2 entity and are now classified either as a subtype of DLBCL, NOS or HGBL, NOS according to their cytomorphological features [8]. However, the Thai Lymphoma Guideline 2022, mentions that in cases where the FISH technique cannot be used, IHC detects MYC and BCL2 expression, referred to as double expression (DE), which may correlate with a poorer prognosis than those without or with single expression [9].

Many previous studies have reported on the biological markers found in patients with DLBCL. However, the investigation of the clinical significance and overall survival outcomes of protein expression via IHC has been less frequently studied in Thai patients with DLBCL, particularly in the northern region of Thailand. This study aims to investigate protein

expression, the clinical impact, and the prognostic role of MYC, BCL2, and Ki-67 expression in Thai DLBCL by IHC technique due to the limitation of FISH, which cannot be generalized in a resource-limited country.

## Materials and methods

### Patients

A total of 100 cases of paraffin-embedded tissue specimens from patients diagnosed with de novo DLBCL and received R-CHOP (rituximab, cyclophosphamide, doxorubicin, vincristine, prednisone) as a standard treatment in the Department of Pathology, Maharaj Nakorn Chiang Mai Hospital (Chiang Mai, Thailand) from January 2018 to December 2019 were enrolled in this study. All cases were reviewed to confirm the diagnosis by one hematopathologist (TD). Clinical data and DLBCL subtypes were retrieved from the lymphoma register form and electronic medical records, spanning the period from June 18, 2021, to December 31, 2022. The information was recorded in a code-linked format, ensuring that individual participants were not directly identified. The retrospective study was conducted with a waiver of the written consent requirement, following the appropriate ethics approval by the Ethics and Research Committee of the Faculty of Medicine, Chiang Mai University [Certificate No. 248/2021 Study code ANA-2563-07765].

### Histopathological study

The patient's lesion was biopsied and immediately preserved in a 10% neutral-buffered formalin solution. The tissues underwent a series of steps including fixing, dehydrating, clearing, and infusion with paraffin wax before being embedded in a paraffin block. Paraffin-embedded tissues were sliced into 3-μm thick sections for histopathological analysis. These sections were subsequently subjected to staining with hematoxylin and eosin (H&E) using standard histological laboratory methods. The histopathological features of the specimens were assessed by experienced pathologists.

### Immunohistochemical study

Sections of 3-μm thick, formalin-fixed, paraffin-embedded tissue were cut and positioned on Superfrost Plus microscope slides. These slides then underwent a one-hour heating procedure at 60 °C within a dry oven, serving to improve tissue adhesion and soften the paraffin. The immunohistochemistry procedure was conducted using a Ventana BenchMark ULTRA autostainer, following the established protocol. In summary, the sections experienced deparaffinization, rehydration, and antigen retrieval using CC1 (prediluted, pH 8.0) antigen retrieval solution by Ventana. Primary antibodies were introduced and allowed to interact with the sections, following the recommended dilution from the manufacturer. The following primary antibodies were used: monoclonal rabbit anti-human MYC (clone Y69, Epitomics, USA, ready-to-use), monoclonal mouse antibody Anti-Human BCL2 (clone 124, Dako, USA, ready-to-use), and monoclonal Mouse Anti-Human Ki-67 (clone MIB-1, Dako, USA, 1:200 dilution) for detecting MYC, BCL2 and Ki-67, respectively. The incubation time was 16 minutes for MYC and 32 minutes for BCL2 and Ki-67. The visualization process was carried out using the Ultraview universal DAB IHC detection kit, which was followed by counterstaining through the application of hematoxylin and bluing solution. The slides were gently cleaned, and dehydrated in a series of graded ethanol and xylene. Finally, the slides were mounted with mounting media onto microscope slides.

### Image analysis

MYC, BCL2, and Ki-67 slides were scanned for each case at the maximum magnification (40X) using Aperio Digital Pathology Slide Scanner. The Digital slides were inspected with Aperio ImageScope Ver.12.4.6.5003 (Leica Biosystem). The region of interest (ROI) was manually selected within the scanned image from the whole sections in each case at x10 magnification and ten fields with an equal area were selected for the analysis at 40X magnification. The image analysis was performed using Membrane Algorithm v.9 for BCL2 expression and Nuclear Algorithm v.9 for MYC and Ki-67 expression. The Cut-off used for positive expression was $\geq 40\%$, $\geq 50\%$, and $\geq 70\%$ for MYC [2], BCL2 [7] and Ki-67 [10], respectively.

### Statistical analysis

The frequencies of each protein expression were counted and calculated in percentages. Fisher's exact test or the chi-square test was conducted to analyze the significance of the association between protein expression and categorical variable parameters, such as sex, and IPI. For survival analysis, OS was defined from the date of receiving specimens to the date of death due to any cause or to the date of the last follow-up. PFS was defined as the elapsed from receiving specimens to tumor progression or death from any cause or the last follow-up with the disease in at least partial remission. The survival curves were generated by the Kaplan-Meier method and using a log-rank test to compare OS and PFS. Univariate and multivariate analyses were performed using Cox proportional-hazards regression. The *P*-value was considered a statistically significant difference when less than 0.05.

## Results

### Patient characteristics

One hundred FFPE tissues of DLBCL patients were included in this study. The clinical characteristics of 100 patients are summarized in Table 1. The median age at diagnosis was 62 years (range, 26–89 years) with 48% of male patients. According to Han's algorithm, there were 21 cases of the GCB subtype and 79 cases with the non-GCB subtype of DLBCL.

### Immunohistochemical results

In the entire 100 cases of DLBCL, 20 cases (20%) displayed MYC-positive tumor cells, and 57 cases (57%) were positive for BCL2 expression. One case (1%) showed double expression with MYC and BCL2. Seven cases (7%) showed double expression with MYC and BCL6. Twelve cases (12%) were triple expressed.

MYC protein expression was notably nuclear staining in all cases (Fig 1A and 1B) and associated with extranodal involvement (*p* = 0.004; Table 2) and Ki-67 expression (*p* = 0.021; Table 2). There were no significant differences in age, sex, COO based on Han's algorithm, Ann Arbor stage, LDH level, performance status, and IPI risk group observed with MYC expression status (Table 2).

BCL2 protein expression was membranous staining (Fig 1C and 1D). The cases with non-GCB subtype and extranodal involvement had BCL2-positive more frequently than the remaining cases (*p* = 0.049 and *p* = 0.026, respectively; Table 2). The cases with BCL2-positive were not associated with age, Ann Arbor stage, LDH level, performance status, IPI risk group, and Ki-67 expression. However, the cases with BCL2-positive tended to correlate with male sex (*p* = 0.061; Table 2).

Ki-67 expression showed nuclear staining (Fig 1E and 1F). Ki-67 expression was associated with MYC expression (*p* = 0.021; Table 2).

**Table 1. Characteristics of DLBCL patients.**

| Characteristics (n = 100) | n (%) |
|---|---|
| **Age** | |
| >60 years | 55 (55) |
| **Sex** | |
| Male | 48 (48) |
| **Cell-of-origin** | |
| GCB | 21 (21) |
| Non-GCB | 79 (79) |
| **Extranodal involvement** | |
| ≥1 | 67 (67) |
| **Ann Arbor stage** | |
| I–II | 40 (40) |
| III–IV | 60 (60) |
| **LDH** | |
| Normal | 28 (28) |
| High | 72 (72) |
| **ECOG performance status** | |
| 0–1 | 60 (60) |
| 2–4 | 40 (40) |
| **IPI risk score** | |
| Low (0–3) | 36 (36) |
| High (>3) | 64 (64) |
| **MYC expression** | |
| Negative | 80 (80) |
| Positive | 20 (20) |
| **BCL2 expression** | |
| Negative | 43 (43) |
| Positive | 57 (57) |
| **Ki-67 expression** | |
| <70% | 81 (81) |
| ≥70% | 19 (19) |
| **Double/Triple expressors** | |
| MYC/BCL2 | 1 (1) |
| MYC/BCL6 | 7 (7) |
| MYC/BCL2/BCL6 | 12 (12) |

## Survival analysis

Median OS and PFS were 23.1 months (range, 0.2–56.3 months) and 17.2 months (range, 0.2–56.3 months), respectively. The 3-year OS was 52% and the 3-year PFS was 46%.

Since the present study used a cut-off ≥40% protein expression for MYC positivity according to the previous study [2], the cases with MYC-positive had similar survival outcomes compared to MYC-negative with three-year OS (48% vs 51%; $p = 0.636$) and three-year PFS (48% vs 45%; $p = 0.523$) (Fig 2A and 2B).

Based on the data published previously [7], this study used ≥50% protein expression for BCL2 positivity. There was no marked significant difference in three-year OS (53% vs. 42%; $p = 0.759$) and 3-year PFS (43% vs. 48%; $p = 0.915$) between the cases with BCL2-positive and negative (Fig 2C and 2D).

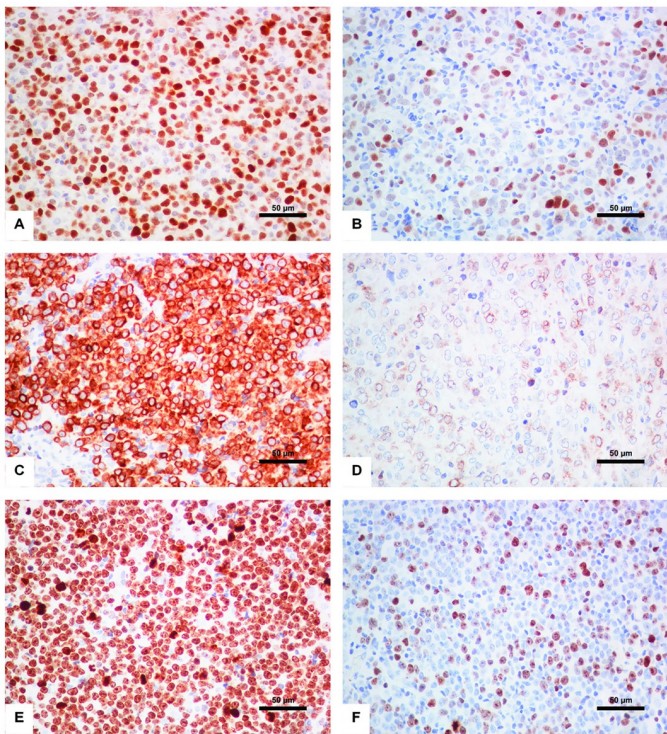

**Fig 1. Immunohistochemical analysis of MYC, BCL2, and Ki-67 expression in DLBCL (A) MYC protein positive (X40); (B) MYC protein negative (X40); (C) BCL2 protein positive (X40); (D) BCL2 protein negative (X40); (E) High Ki-67 expression (X40); (F) Low Ki-67 expression (X40).**

For Ki-67 expression, this study found similar three-year OS (38% vs. 54%; $p = 0.401$) and three-year PFS (43% vs. 45%; $p = 0.628$) in the cases with high Ki-67 and low Ki-67 expression by the cut-off point of $\geq 70\%$ (Fig 2E and 2F).

In addition, the cases with $\geq 60$ years old had shorter three-year OS (37% vs 70%; $p = 0.001$) than those the cases with $< 60$ years old (Fig 3A). The cases with high IPI score (4–8) had shorter three-year OS (41% vs 68%; $p = 0.012$) than those the cases with low IPI scores (0–3) (Fig 3B). With regards to ECOG scores, the low ECOG scores were significantly associated with longer three-year OS (74% vs 17%; $p < 0.001$) compared with the cases with high ECOG scores and likewise three-year PFS (51% vs 30%; $p = 0.037$) (Fig 3C and 3D).

Double expressor and triple expressor DLBCL cases were then analyzed, and the results showed that the case with MYC-positive/BCL2-positive is significantly inferior only in three-year OS compared with all others (0% vs 53%; $p = 0.020$) (Fig 4) but not in three-year PFS. This study observed no mark differences in three-year OS and three-year PFS in both cases with double expression MYC-positive/BCL6-positive and triple expression.

Univariable and multivariable analysis for PFS and OS, age, and poor ECOG performance status demonstrated the poor prognostic effect only in OS after adjusting for the presence of the additional factors of gender, the status of biopsy, stage III-IV, extranodal involvement, high LDH level, IPI risk score, cell-of-origin, MYC expression, BCL2 expression, Ki-67 expression and double or triple expression (Tables 3 and 4).

**Table 2. Comparison of protein expression and clinical appearance.**

| | MYC expression | | P-value | BCL2 expression | | P-value | Ki-67 expression | | P-value |
|---|---|---|---|---|---|---|---|---|---|
| | <40% | ≥40% | | <50% | ≥50% | | <70% | ≥70% | |
| n (%) | 80 (80.0) | 20 (20.0) | | 43 (43.0) | 57 (57.0) | | 81 (81.0) | 19 (19.0) | |
| **Age (years)** | | | | | | | | | |
| <60 | 35 (43.8) | 10 (50.0) | 0.615 | 17 (39.5) | 28 (49.1) | 0.340 | 36 (44.4) | 9 (47.4) | 0.818 |
| ≥60 | 45 (56.3) | 10 (50.0) | | 26 (60.5) | 29 (50.9) | | 45 (55.6) | 10 (52.6) | |
| **Sex** | | | | | | | | | |
| Male | 39 (48.8) | 9 (45.0) | 0.764 | 16 (37.2) | 32 (56.1) | 0.061 | 40 (49.4) | 8 (42.1) | 0.568 |
| Female | 41 (51.2) | 11 (55.0) | | 27 (62.8) | 25 (43.9) | | 41 (50.6) | 11 (57.9) | |
| **Cell-of-origin** | | | | | | | | | |
| GCB | 19 (23.8) | 2 (10.0) | 0.230 | 13 (30.2) | 8 (14.0) | **0.049** | 18 (22.2) | 3 (15.8) | 0.756 |
| Non-GCB | 61 (76.2) | 18 (90.0) | | 30 (69.8) | 49 (86.0) | | 63 (77.8) | 16 (84.2) | |
| **Extranodal involvement** | | | | | | | | | |
| Nodal | 21 (26.2) | 12 (60.0) | **0.004** | 9 (20.9) | 24 (42.1) | **0.026** | 27 (33.3) | 6 (31.6) | 0.884 |
| ≥1 | 59 (73.8) | 8 (40.0) | | 34 (79.1) | 33 (57.9) | | 54 (66.7) | 13 (68.4) | |
| **Ann Arbor stage** | | | | | | | | | |
| I–II | 32 (40.0) | 8 (40.0) | 1.000 | 15 (34.9) | 25 (43.9) | 0.364 | 32 (39.5) | 8 (42.1) | 0.835 |
| III–IV | 48 (60.0) | 12 (60.0) | | 28 (65.1) | 32 (56.1) | | 49) 60.5) | 11 (57.9) | |
| **LDH** | | | | | | | | | |
| Normal | 25 (31.2) | 3 (15.0) | 0.148 | 13 (30.2) | 15 (26.3) | 0.666 | 23 (28.4) | 5 (26.3) | 0.856 |
| High | 55 (68.8) | 17 (85.0) | | 30 (69.8) | 42 (73.7) | | 58 (71.6) | 14 (73.7) | |
| **ECOG score** | | | | | | | | | |
| 0–1 | 48 (60.0) | 12 (60.0) | 1.000 | 24 (55.8) | 36 (63.2) | 0.458 | 50 (61.7) | 10 (52.6) | 0.466 |
| 2–4 | 32 (40.0) | 8 (40.0) | | 19 (44.2) | 21 (36.8) | | 31 (38.3) | 9 (47.4) | |
| **IPI risk score** | | | | | | | | | |
| Low (0–3) | 27 (33.8) | 9 (45.0) | 0.349 | 13 (30.2) | 23 (40.4) | 0.297 | 29 (35.8) | 7 (36.8) | 0.932 |
| High (>3) | 53 (66.3) | 11 (55.0) | | 30 (69.8) | 34 (59.6) | | 52 (64.2) | 12 (63.2) | |
| **MYC expression** | | | | | | | | | |
| <40% | - | - | - | 36 (83.7) | 44 (77.2) | 0.419 | 69 (85.2) | 11 (57.9) | **0.021** |
| ≥40% | - | - | | 7 (16.3) | 13 (22.8) | | 12 (14.8) | 8 (42.1) | |
| **BCL2 expression** | | | | | | | | | |
| <50% | 36 (45.0) | 7 (35.0) | 0.419 | - | - | - | 36 (44.4) | 7 (36.8) | 0.547 |
| ≥50% | 44 (55.0) | 13 (65.0) | | - | - | | 45 (55.6) | 12 (63.2) | |
| **Ki-67 expression** | | | | | | | | | |
| <70% | 69 (86.3) | 12 (60.0) | **0.021** | 36 (83.7) | 45 (78.9) | 0.547 | - | - | - |
| ≥70% | 11 (13.8) | 8 (40.0) | | 7 (16.3) | 12 (21.1) | | - | - | |

## Discussion

Diffuse large B-cell lymphoma (DLBCL) is the most common aggressive B-cell lymphoma with marked biologic heterogeneity. Research on the role of proteins MYC, BCL2, and Ki-67 in lymphomagenesis is vast. This present study stated the correlation between the protein expression and clinical characteristics according to the cut-off point reported by the previous study and further analyzed different cut-off values regarding exploring the optimal cut-off point for Thai DLBCL. In addition, the three-year OS and three-year PFS were also investigated. Due to the limitations and the study's scope, FISH results are not included in this study; thus, protein expression is assessed via the IHC technique, precluding classification according

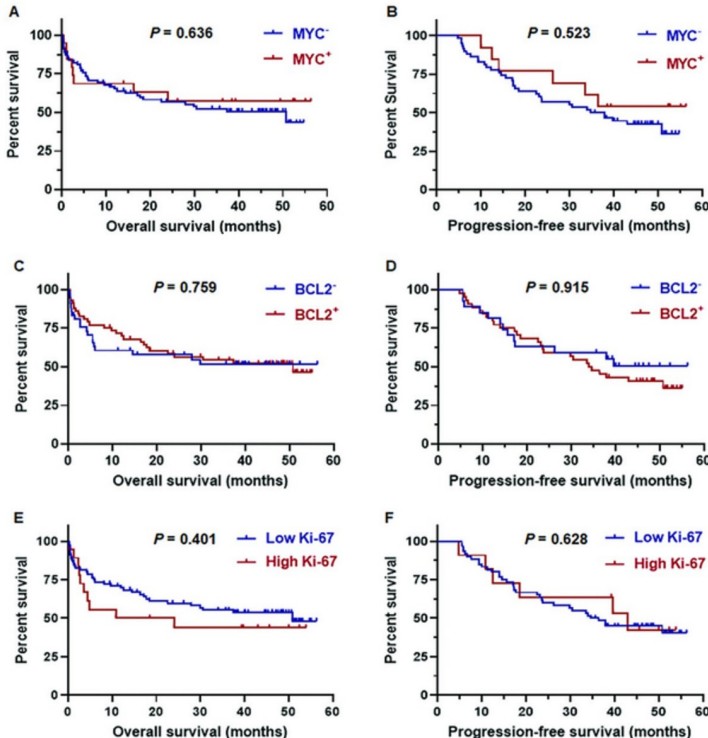

**Fig 2. Overall survival and progression-free survival of MYC, BCL2, and Ki-67 expression in DLBCL by Kaplan-Meier analysis.**

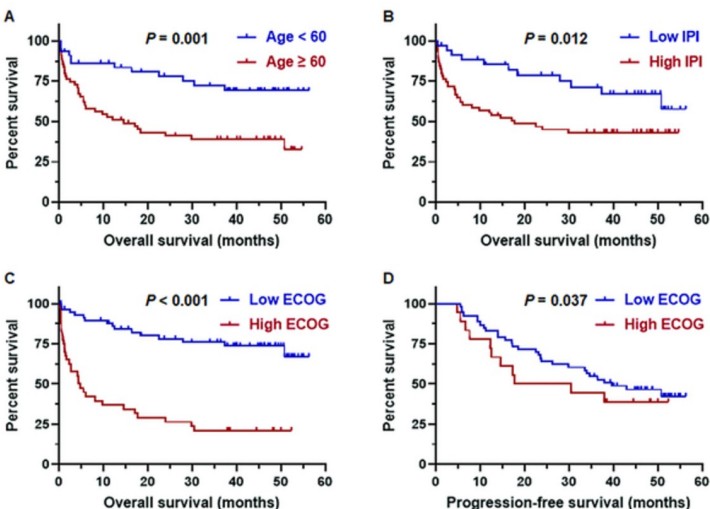

**Fig 3. Overall survival and progression-free survival of age, IPI risk score, and ECOG score in DLBCL by Kaplan-Meier analysis.**

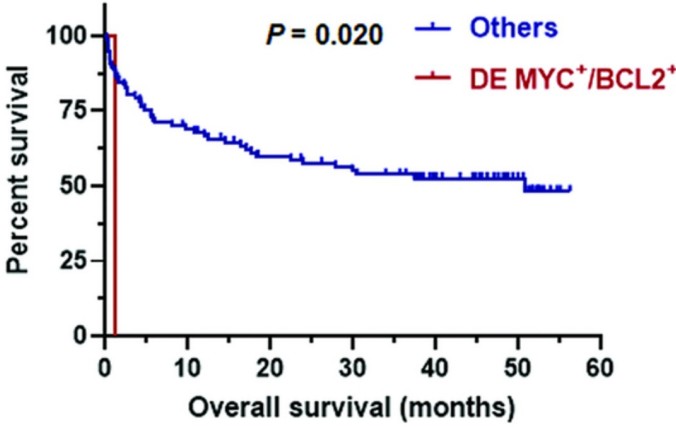

**Fig 4. Overall survival of double expressor in DLBCL by Kaplan-Meier analysis.**

**Table 3. Univariable and multivariable analysis for progression-free survival.**

| Covariates | Values | Cases | Univariable analysis | | Multivariable analysis | |
|---|---|---|---|---|---|---|
| | | | HR (95% CI) | *P*-value | HR (95% CI) | *P*-value |
| Age | < 60 years | 45 | Reference | | Reference | |
| | ≥ 60 years | 55 | 1.47 (0.83–2.61) | 0.183 | 1.36 (0.76–2.42) | 0.299 |
| Sex | Female | 48 | Reference | | | |
| | Male | 52 | 1.07 (0.60–1.91) | 0.801 | - | - |
| Status at biopsy | New cases | 88 | Reference | | | |
| | Relapse cases | 12 | 1.41 (0.63–3.15) | 0.400 | - | - |
| Stage | I–II | 40 | Reference | | | |
| | III–IV | 60 | 1.44 (0.79–2.61) | 0.225 | - | - |
| Extranodal involvement | No | 33 | Reference | | | |
| | Yes | 67 | 1.13 (0.61–2.08) | 0.694 | - | - |
| LDH | Normal | 28 | Reference | | | |
| | High | 72 | 0.83 (0.46–1.53) | 0.566 | - | - |
| ECOG PS | 0–2 | 58 | Reference | | Reference | |
| | 3–4 | 33 | 1.86 (1.02–3.37) | **0.040** | 1.76 (0.96–3.22) | 0.065 |
| IPI risk score | 0–2 | 37 | Reference | | | |
| | 3–5 | 63 | 1.02 (0.58–1.81) | 0.925 | - | - |
| Cell-of-origin | GCB | 21 | Reference | | | |
| | non-GCB | 79 | 0.84 (0.44–1.63) | 0.622 | - | - |
| MYC expression | negative | 80 | Reference | | | |
| | positive | 20 | 0.78 (0.36–1.67) | 0.525 | - | - |
| BCL2 expression | negative | 43 | Reference | | | |
| | positive | 57 | 1.03 (0.57–1.85) | 0.915 | - | - |
| Ki-67 expression | < 70% | 81 | Reference | | | |
| | ≥ 70% | 19 | 1.19 (0.57–2.47) | 0.628 | - | - |
| Double/Triple expressor | No | 80 | Reference | | | |
| | Yes | 20 | 0.78 (0.36–1.67) | 0.525 | - | - |

**Table 4. Univariable and multivariable analysis for overall survival.**

| Covariates | Values | Cases | Univariable analysis | | Multivariable analysis | |
|---|---|---|---|---|---|---|
| | | | HR (95% CI) | *P*-value | HR (95% CI) | *P*-value |
| Age | < 60 years | 45 | Reference | | Reference | |
| | ≥ 60 years | 55 | 2.83 (1.46–5.49) | **0.002** | 2.18 (1.08–4.38) | **0.029** |
| Sex | Female | 48 | Reference | | | |
| | Male | 52 | 0.71 (0.39–1.27) | 0.250 | - | - |
| Status at biopsy | New cases | 88 | Reference | | | |
| | Relapse cases | 12 | 1.55 (0.72–3.32) | 0.260 | - | - |
| Stage | I–II | 40 | Reference | | | |
| | III–IV | 60 | 1.07 (0.48–2.39) | 0.870 | - | - |
| Extranodal involvement | No | 33 | Reference | | | |
| | Yes | 67 | 1.04 (0.56–1.93) | 0.887 | - | - |
| LDH | Normal | 28 | Reference | | | |
| | High | 72 | 1.50 (0.74–3.03) | 0.253 | - | - |
| ECOG PS | 0–2 | 58 | Reference | | Reference | |
| | 3–4 | 33 | 5.40 (2.88–10.11) | **<0.001** | 5.77 (2.53–13.15) | **<0.001** |
| IPI risk score | 0–2 | 37 | Reference | | Reference | |
| | 3–5 | 63 | 2.32 (1.17–4.57) | **0.015** | 0.65 (0.25–1.65) | **0.372** |
| Cell-of-origin | GCB | 21 | Reference | | | |
| | non-GCB | 79 | 1.20 (0.58–2.49) | 0.617 | - | - |
| MYC expression | negative | 80 | Reference | | | |
| | positive | 20 | 0.83 (0.38–1.78) | 0.637 | - | - |
| BCL2 expression | negative | 43 | Reference | | | |
| | positive | 57 | 0.91 (0.50–1.64) | 0.760 | - | - |
| Ki-67 expression | < 70% | 81 | Reference | | | |
| | ≥ 70% | 19 | 1.34 (0.66–2.72) | 0.403 | - | - |
| Double/Triple expressor | No | 80 | Reference | | | |
| | Yes | 20 | 0.83 (0.38–1.78) | 0.637 | - | - |

to WHO-HAEM5 [8] due to FISH limitations. Nonetheless, these findings provide valuable prognostic insights, particularly in resource-limited countries.

Previous studies have reported that many clinical factors were found to be insubstantial [11]. However, this study discovered a correlation between protein expression and clinical outcomes. While DLBCL can affect individuals across various age groups and sexes, the mean age of the cases in this study was 62 years (range, 26–89 years) and the disease occurs equally in both sexes.

The MYC protein acts as a transcription factor, playing a role in regulating cell cycle processes such as cell proliferation, growth, DNA replication, and protein synthesis. In approximately one-third of DLBCL cases, there is an observed increase in the expression of MYC protein [11]. This suggests that differences in MYC protein levels could be used as a prognostic biomarker. The cut-off value for defining positivity for protein expression varies among studies and depends on their aims. In the present study, MYC-positive expression by the cut-off point of ≥ 40% was found in 20 out of 100 cases (20%) and associated with extranodal involvement. Castillo and colleagues reported that approximately one-third of cases with DLBCL display extranodal involvement, which gastrointestinal, pulmonary, and hepatobiliary/pancreatic sites were notably worse than those for nodal sites [12]. Further studies should investigate the correlation between protein biomarkers and extranodal involvement in Thai patients with

DLBCL to elucidate the clinical impact and enable proper management. MYC expression was also found to be associated with Ki-67 expression at a cut-off point of $\geq$ 70%, which is in line with the previous studies [13]. Moreover, this study further analyzed MYC expression at different cut-off values, The three-year OS in the MYC-positive group was significantly lower than in the MYC-negative group (0% vs 53%; $p$ = 0.001) by the cut-off point of $\geq$ 80% which revealed similar tendency being observed in three-year PFS (0% vs 46%; $p$ = < 0.001). Most findings demonstrated MYC expression as a prognostic factor of poor survival, in agreement with this finding but with different cut-off points [2, 11, 14, 15].

The BCL2 protein prevents programmed cell death in cells, and its overexpression can occur through gene amplification or translocation in DLBCL. However, there is still no consensus on the clinical significance of BCL2 protein expression in DLBCL, and previous studies have produced conflicting results on the impact of BCL2 overexpression on survival. Moreover, the prognostic value of BCL2 protein overexpression differs between the GCB and ABC subtypes [16, 17]. This study showed that BCL2 protein expression has no significant impact on three-year OS and PFS by the cut-off point $\geq$ 50%. Furthermore, the analysis using the Kaplan-Meier method in this study demonstrated a trend towards lower three-year PFS in patients with BCL2-positive cases compared to those with BCL2-negative cases using the determining BCL2 positivity at $\geq$ 40% but the difference was not statistically significant (39% vs 56%; $p$ = 0.190). Nevertheless, the overexpression of BCL2 was associated with the non-GCB subtype, which was in accord with prior studies [18]. Iqbal and colleagues stated that the inconsistent findings regarding the predictive value of BCL2 expression in DLBCL in various studies can be attributed to several factors. Firstly, the heterogeneity of the DLBCL cases studied, including the different proportions of GCB and ABC-DLBCL cases. Secondly, the patient population may have different risk factors apart from the distinction of DLBCL subtype. Thirdly, the variations in the management of the patients could also be a factor. Fourthly, technical factors affecting immunostaining could play a role. Finally, the experience and subjectivity of the pathologist scoring the cases may contribute to the conflicting results observed in the literature [16].

Ki-67 is a nuclear marker that indicates not only the level of lymphoproliferative activity in non-Hodgkin's lymphoma but also in various tumors. It is routinely used as a measure of proliferation and has potential prognostic value for various subtypes of DLBCL. Although there have been many attempts to correlate its expression with other biological markers, as well as with clinical and therapeutic outcomes, no conclusive findings have been reported [19, 20]. The lack of a consensus on the definition of low and high Ki-67 expression may contribute to the inconsistencies in the results of studies using this marker. One possible explanation for these discrepancies could be the use of different threshold values to define Ki-67 expression status. The present study further analyzed Ki-67 expression at different cut-off values, a cut-off value of $\geq$ 50% was found to be the most effective in discriminating Ki-67 expression and was also identified as a reliable predictor of survival outcome in patients with DLBCL (three-year OS; 60% vs 34%; $p$ = 0.026).

Although DLBCL can affect individuals of all ages, it is more common in older adults and associated with poor general health status (ECOG performance status score of 2 or greater) [21]. IPI is a method that involves a simple scoring system that combines different biological features such as age, LDH level, Ann Arbor stage, extranodal site, and ECOG performance status. IPI is effective in predicting patient survival. The results of this study indicated that the prognostic value of IPI is particularly pronounced in high IPI scores, which have been found to have a worse three-year OS than those with a low IPI score.

DEL is associated with other high-risk clinical parameters, such as high Ki-67 expression, high ECOG score, and high IPI score. MYC and BCL2 expression, and often displays an

independent adverse prognosis compared to other DLBCL cases [22]. The only identified DEL case in this study was an elderly individual (73 years old) who had at least an IPI score of 4. Statistical analysis showed that DEL (MYC-positive/BCL2-positive type) is associated with a poor three-year OS. However, due to the limited number of cases, this study cannot provide an estimate of independent risk factors for survival outcomes. Green and colleagues reported that there was no notable variation in the survival rate between DEL and DHL, indicating that DEL behaves biologically in a similar manner as DHL. This implies that "double expression" is connected to a poor prognosis of lymphoma [6].

Nowadays, various diagnostic approaches are available for DLBCL. The Hans algorithm, which employs an IHC-based technique, remains a fundamental aspect of the initial assessment of DLBCL, eliminating the necessity for gene expression studies [23, 24]. IHC is widely available in most pathology laboratories, enabling the simultaneous examination of morphology and immunophenotypic characteristics. Additionally, it is generally more cost-effective compared to certain other molecular techniques. Mohammed and colleagues identified the potential of using IHC rather than RT-PCR in developing countries due to the statistically significant correlation observed between mRNA levels of MYC and BCL2 and their protein expression ($p<0.001$). These findings suggest that there was no difference in disease prognosis between mRNA levels and protein expression levels [25]. Furthermore, IHC should be conducted for MYC and BCL2 to detect DEL-DLBCL, which has been linked to a relatively unfavorable prognosis and utilized to eliminate unnecessary cytogenetic FISH testing. However, given that 80% of DHL cases have DEL (MYC/BCL2) [24], and because IHC cannot detect gene rearrangements such as translocations, cytogenetic FISH testing for cases presenting with these markers would aid in classifying them according to the recent WHO-HAEM5 classification [8] and identifying DHL cases with even poorer outcomes and the need for a more intensive treatment plan. Swerdlow [22] stated that DEL is not equivalent to DHL/THL and is more prevalent than DHL, which is found in approximately 19% to 34% of DLBCL cases [6, 26–28] and with around 20% of DEL being DHL [6].

The univariate analysis for PFS revealed that only poor ECOG performance status demonstrated a poor prognostic effect. Subsequently, this study adjusted the variable age in the multivariate analysis, and the results showed that both age and ECOG performance status no longer had a prognostic effect on PFS. Regarding the univariable analysis for OS, age, ECOG performance status, and IPI risk group were associated with OS. However, after adjusting for multivariable factors, age, and ECOG performance status remained prognostic factors for OS.

In conclusion, MYC, BCL2, and Ki-67 expression can be used as prognostic biomarkers. However, Since the prognostic significance of each biomarker varies depending on the cut-off value used, it's important to determine the appropriate cut-off level for each biomarker based on the clinical results of the individual laboratories. Further studies should investigate the correlation between protein biomarkers and extranodal involvement in Thai patients with DLBCL to elucidate the clinical impact and enable proper management.

## Acknowledgments

The authors would like to thank Ms. Lakkana Eianleng and Ms. Lamaiporn Peerapapong for technical assistance and Mrs. Rochana Phuackchantuck for statistical consultations.

## Author Contributions

**Conceptualization:** Phuttirak Yimpak, Kanokkan Bumroongkit, Adisak Tantiworawit, Thanawat Rattanathammethee, Sirinda Aungsuchawan, Teerada Daroontum.

**Data curation:** Phuttirak Yimpak, Kanokkan Bumroongkit, Adisak Tantiworawit, Thanawat Rattanathammethee, Sirinda Aungsuchawan, Teerada Daroontum.

**Formal analysis:** Phuttirak Yimpak, Kanokkan Bumroongkit, Adisak Tantiworawit, Thanawat Rattanathammethee, Teerada Daroontum.

**Funding acquisition:** Phuttirak Yimpak, Kanokkan Bumroongkit.

**Investigation:** Phuttirak Yimpak, Kanokkan Bumroongkit, Adisak Tantiworawit, Thanawat Rattanathammethee, Teerada Daroontum.

**Methodology:** Phuttirak Yimpak, Teerada Daroontum.

**Project administration:** Phuttirak Yimpak, Kanokkan Bumroongkit, Teerada Daroontum.

**Resources:** Adisak Tantiworawit, Thanawat Rattanathammethee, Teerada Daroontum.

**Supervision:** Kanokkan Bumroongkit, Teerada Daroontum.

**Validation:** Phuttirak Yimpak, Kanokkan Bumroongkit, Adisak Tantiworawit, Thanawat Rattanathammethee, Teerada Daroontum.

**Visualization:** Phuttirak Yimpak.

**Writing – original draft:** Phuttirak Yimpak.

**Writing – review & editing:** Phuttirak Yimpak, Kanokkan Bumroongkit, Adisak Tantiworawit, Thanawat Rattanathammethee, Sirinda Aungsuchawan, Teerada Daroontum.

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
