## [Decision Letter · Decision Letter 0]

28 Feb 2024

PONE-D-23-39686Clinical impacts and prognostic role of MYC, BCL2, and Ki-67 expression in patients with diffuse large B-cell lymphomaPLOS ONE

Dear Dr. Daroontum,

Thank you for submitting your manuscript to PLOS ONE. After careful consideration, we feel that it has merit but does not fully meet PLOS ONE’s publication criteria as it currently stands. Therefore, we invite you to submit a revised version of the manuscript that addresses the points raised during the review process. Although the manuscript contains retrospective data; I believe that it is critical to have information about DLBCL from a developing country. Therefore the manuscript needs some changes and suggestions.

We look forward to receiving your revised manuscript.

Kind regards,

Mehmet Baysal

Academic Editor

PLOS ONE

Journal Requirements:

2. Thank you for stating the following financial disclosure: "This work was supported by the Faculty of Medicine Research Fund, Chiang Mai University, Chiang Mai, Thailand [grant numbers ANA-2563-07765, No. 160/2564]."

Reviewers' comments:

Reviewer's Responses to Questions

**Comments to the Author**

1. Is the manuscript technically sound, and do the data support the conclusions?

Reviewer #1: Partly

Reviewer #2: Yes

2. Has the statistical analysis been performed appropriately and rigorously? 

Reviewer #1: No

Reviewer #2: I Don't Know

3. Have the authors made all data underlying the findings in their manuscript fully available?

Reviewer #1: Yes

Reviewer #2: Yes

4. Is the manuscript presented in an intelligible fashion and written in standard English?

Reviewer #1: Yes

Reviewer #2: Yes

5. Review Comments to the Author

Reviewer #1: This is a nice paper, but there are multiple issues of concern.

1. Who reported the final pathology findings (1 or more haematopathologist)?

2. Were all cases de novo DLBCL or was there transformed cases?

3. What did the patient receive for treatment?

4. Usually head and neck localization of DLBCL have worse prognosis, but in your cases, this was opposite? Wha is the explanation for this?

5. Was gene rearangement performed or only protein expression?

6. DEL cases usually have better prognosis than DHL, but in your cases this was also opposite.

7. As authors mentioned in the manuscript, Iqban et al. had shown that in pre-rituximab era that BCL-2 protein over-expression was prognostic only in the non-GCB subtype, but in another report Iqban found the opposite where BCL-2 protein over expression was significantly associated with an inferior outcome in the GCB subtype. BCL2 expression is usually more prominent in GCB subtype. What is the explanation for the opposite findings in Thai patientsí?

8. What is the authors opinion about the difference in prognosis at mRNA and protein expression level?

9. DLBCL patients with either MYC/BCL6 rearrangements or MYC/BCL6 co-expression do not always have poorer prognosis, but high level of MYC expression might have negative effect on survival, therefore its levels should be evaluated simultaneously.

10. Previous reports found that the survival of patients with high Ki-67 was inferior with respect to OS, but not PFS. What does the authors think of this?

Reviewer #2: With only the data source, this study is interesting and promising, coming from a developing country and adding more information regarding the characteristics of DLBCL in the world population. Having more information regarding patient variability from different patient characteristics is undoubtedly valuable. The manuscript is also written in a good flow of thinking thread of the study while offering a more accessible modality for daily use in limited environmental circumstances. Thus, to enhance the study's strength amidst the many molecular analyses utilized regarding DLBCL, I suggest adjusting the title to highlight the appeal of the benefit of what the study offered.

Small technical detail additions in the IHC method, such as the incubation time and dilution for MYC and BCl2, would also help readers interested in repeating the method. Also, additional insights into comparing the limitation of IHC with other well-known molecular analyses in DLBCL diagnosis would help the reader who is still getting familiar with the topic.

Images and graphs are decent; a small note for Table 1 titled "clinical" also contains the result of IHC expression. Although, this is optional to revise.

Thank you for sharing your study.

6. PLOS authors have the option to publish the peer review history of their article (what does this mean?). If published, this will include your full peer review and any attached files.

Reviewer #1: **Yes: **Hussain Alizadeh

Reviewer #2: No

---

## [Author Response · Author response to Decision Letter 0]

29 Mar 2024

We would like to express our gratitude to the reviewers for taking the time to read and provide valuable feedback like this. Below is our reply to your comment.

Has the statistical analysis been performed appropriately and rigorously?

This study had already been conducted and reviewed appropriately, according to the statistician, Mrs. Rochana Phuackchantuck, statistical consultant and research support officer at the Faculty of Medicine, Chiang Mai University. We mentioned in acknowledgement section (Page 23, line 374-376).

Reviewer #1: This is a nice paper, but there are multiple issues of concern.

1. Who reported the final pathology findings (1 or more haematopathologist)?

All the diagnostics had been confirmed by one hematopathologist (TD). We added the statement “All cases were reviewed to confirm the diagnosis by one hematopathologist (TD).” In Materials and Methods section (Page 5, line 99-100).

2. Were all cases de novo DLBCL or was there transformed cases?

All cases are de novo DLBCL. We added the statement “A total of 100 cases of paraffin-embedded tissue specimens from patients diagnosed with de novo DLBCL and received R-CHOP (rituximab, cyclophosphamide, doxorubicin, vincristine, prednisone) as a standard treatment in the Department of Pathology, Maharaj Nakorn Chiang Mai Hospital (Chiang Mai, Thailand) from January 2018 to December 2019 were enrolled in this study.” in Materials and Methods section (Page 5, line 95-97).

3. What did the patient receive for treatment?

All patients received Rituximab + CHOP (R-CHOP regimen) as a standard treatment. We added the statement “A total of 100 cases of paraffin-embedded tissue specimens from patients diagnosed with de novo DLBCL and received R-CHOP (rituximab, cyclophosphamide, doxorubicin, vincristine, prednisone) as a standard treatment in the Department of Pathology, Maharaj Nakorn Chiang Mai Hospital (Chiang Mai, Thailand) from January 2018 to December 2019 were enrolled in this study.” in Materials and Methods section (Page 5, line 95-97).

4. Usually head and neck localization of DLBCL have worse prognosis, but in your cases, this was opposite? Wha is the explanation for this?

According to Castillo et al., 2014., in our previous version, we intend to underscore that extranodal sites typically indicate a worse prognosis. They stated that, based on their research, all extranodal sites affected by DLBCL showed a lower percentage of advanced stage disease at diagnosis compared to lymph node involvement. Similar findings have also been documented in studies conducted in Asia. One possible explanation for this observation is that the presence of extranodal involvement may lead to noticeable early and earlier detection due to symptoms related to tumor mass effect. 

However, to prevent confusion, we have already removed that portion from the discussion section. We deleted the statement “Head and neck sites appeared as independent sites with a better prognosis when compared to nodal sites”.

5. Was gene rearangement performed or only protein expression?

This study only evaluated protein expression.

6. DEL cases usually have better prognosis than DHL, but in your cases this was also opposite.

Thank you for your thoughtful suggestion. However, our study did not evaluate instances of DHL. Consequently, there were no findings accessible regarding the comparison of survival outcomes between DEL and DHL.

7. As authors mentioned in the manuscript, Iqban et al. had shown that in pre-rituximab era that BCL-2 protein over-expression was prognostic only in the non-GCB subtype, but in another report Iqban found the opposite where BCL-2 protein over expression was significantly associated with an inferior outcome in the GCB subtype. BCL2 expression is usually more prominent in GCB subtype. What is the explanation for the opposite findings in Thai patientsí?

Our results were similar to the prior studies from Punnoose and colleagues. We stated about BCL2 protein expression in discussion section mentioned in page 12 line 189-192 and page 19 line 294-296).

Based on our study, we identified a subset of patients with BCL2 expression. A possible explanation is that the majority of DLBCL patients were non-GCB (n = 79 in non-GCB vs. 21 in GCB). This may imply that the difference in proportion of GCB and non-GCB leads to statistical significance in BCL2 expression in this study.

8. What is the authors opinion about the difference in prognosis at mRNA and protein expression level?

We added the statement “Mohammed and colleagues identified the potential of using IHC rather than RT-PCR in developing countries due to the statistically significant correlation observed between mRNA levels of MYC and BCL2 and their protein expression (p<0.001). These findings suggest that there was no difference in disease prognosis between mRNA levels and protein expression levels” in discussion section (Page 21, line 345-350).

This study led the authors to conclude that there was no difference in disease prognosis between mRNA levels and protein expression levels. 

9. DLBCL patients with either MYC/BCL6 rearrangements or MYC/BCL6 co-expression do not always have poorer prognosis, but high level of MYC expression might have negative effect on survival, therefore its levels should be evaluated simultaneously.

We agree with reviewer that MYC/BCL6 do not always have independent prognostic. Ye and colleagues (2015) stated that DLBCL patients with MYC/BCL6 co-expression showed a significantly poorer survival than DLBCL patients without MYC/BCL6 co-expression. However, this prognostic effect was significant only in the presence of DLBCL with MYC/BCL2 co-expression. The isolated MYC+BCL2-BCL6+ (from all MYC+BCL6+) subgroup had significantly better patient survival compared with the MYC+BCL2+BCL6- (from all MYC+BCL2+) subgroup. 

Additionally, this study observed no mark differences in three-year OS and three-year PFS in double expression MYC-positive/BCL6-positive. Due to the aim of this study, we only performed IHC analysis on three proteins: MYC, BCL2, and Ki-67. For BCL6, we collected the result to classify subgroups according to Han's algorithm. This limitation means we cannot provide the actual level of expression of BCL6. However, we might evaluate MYC and BCL6 simultaneously in further studies.

10. Previous reports found that the survival of patients with high Ki-67 was inferior with respect to OS, but not PFS. What does the authors think of this?

In OS analysis, if patients die before achieving remission, their deaths are considered events and are included in the analysis. This contributes to the overall survival analysis, as the occurrence of death is the primary endpoint.

In PFS analysis, if patients die before achieving remission, their deaths are considered events only if they are related to disease progression. Deaths unrelated to disease progression are censored, meaning they are not considered as progression events. Therefore, if a patient dies before achieving remission but the death is not due to disease progression, it may not be counted as an event in PFS analysis, thus not affecting its significance. 

In summary, deaths occurring before remission contribute to the significance of OS analysis but may not significantly impact the significance of PFS analysis, as PFS primarily focuses on disease progression events. For this issue, it's important to consider that OS can be influenced by various factors, including treatment-related mortality and non-lymphoma-related causes of death. Additionally, due to data limitations and challenges in accurately identifying causes of death. This interesting data should be further investigated in the future study.

Reviewer #2: With only the data source, this study is interesting and promising, coming from a developing country and adding more information regarding the characteristics of DLBCL in the world population. Having more information regarding patient variability from different patient characteristics is undoubtedly valuable. The manuscript is also written in a good flow of thinking thread of the study while offering a more accessible modality for daily use in limited environmental circumstances. Thus, to enhance the study's strength amidst the many molecular analyses utilized regarding DLBCL, I suggest adjusting the title to highlight the appeal of the benefit of what the study offered.

Small technical detail additions in the IHC method, such as the incubation time and dilution for MYC and BCl2, would also help readers interested in repeating the method. Also, additional insights into comparing the limitation of IHC with other well-known molecular analyses in DLBCL diagnosis would help the reader who is still getting familiar with the topic.

Images and graphs are decent; a small note for Table 1 titled "clinical" also contains the result of IHC expression. Although, this is optional to revise.

Thank you for sharing your study.

We sincerely appreciate your thoughtful suggestions. Your insights are highly valued, and as a result, we would like to address all the issues as follows:

We have revised our title based on your feedback to read: “Immunohistochemistry-based investigation of MYC, BCL2, and Ki-67 protein expression and their clinical impact in diffuse large B-cell lymphoma”.

For small technical detail additions in the IHC method, such as the incubation time and dilution for MYC and BCl2, we have stated that issues in material and method part (page 7 line 129-130). The antibodies for MYC and BCL2 were used as ready-to-use solutions, while Ki-67 was diluted to a concentration of 1:200.

In regarding additional insights into comparing the limitation of IHC with other well-known molecular analyses in DLBCL diagnosis, we have added the statement “Nowadays, various diagnostic approaches are available for DLBCL. The Hans algorithm, which employs an IHC-based technique, remains a fundamental aspect of the initial assessment of DLBCL, eliminating the necessity for gene expression studies [24, 25]. IHC is wildly available in most pathology laboratories, enabling the simultaneous examination of morphology and immunophenotypic characteristics. Additionally, it is generally more cost-effective compared to certain other molecular techniques. Mohammed and colleagues identified the potential of using IHC rather than RT-PCR in developing countries due to the statistically significant correlation observed between mRNA levels of MYC and BCL2 and their protein expression (p<0.001). These findings suggest that there was no difference in disease prognosis between mRNA levels and protein expression levels [26]. Furthermore, IHC should be conducted for MYC and BCL2 to detect DEL-DLBCL, which has been linked to a relatively unfavorable prognosis and utilized to eliminate unnecessary cytogenetic FISH testing. However, given that 80% of DHL cases have DEL (MYC/BCL2) [25], cytogenetic FISH testing for cases presenting with these markers would aid in identifying DHL cases with even poorer outcomes and the need for more intensive treatment plan.” in discussion part (page 21-22 line 339-355).

Finally, we have deleted the word “clinical” for Table 1 titled already.

---

## [Decision Letter · Decision Letter 1]

10 May 2024

PONE-D-23-39686R1Immunohistochemistry-based investigation of MYC, BCL2, and Ki-67 protein expression and their clinical impact in diffuse large B-cell lymphomaPLOS ONE

Dear Dr. Daroontum,

Thank you for submitting your manuscript to PLOS ONE. After careful consideration, we feel that it has merit but does not fully meet PLOS ONE’s publication criteria as it currently stands. Therefore, we invite you to submit a revised version of the manuscript that addresses the points raised during the review process.

The study group is small and diverse, collected from a single center without a correlation to the "gold standard" FISH.I thnk a broad revision could overcome this problematic errors. Please consider  adding the nation of origin to the title and emphasizing the limited scope of the study in the introduction. Referring to the most recent version is necessary to indicate the study's restrictions at the time of publishing.

We look forward to receiving your revised manuscript.

Kind regards,

Mehmet Baysal

Academic Editor

PLOS ONE

Reviewers' comments:

Reviewer's Responses to Questions

**Comments to the Author**

1. If the authors have adequately addressed your comments raised in a previous round of review and you feel that this manuscript is now acceptable for publication, you may indicate that here to bypass the “Comments to the Author” section, enter your conflict of interest statement in the “Confidential to Editor” section, and submit your "Accept" recommendation.

Reviewer #2: All comments have been addressed

Reviewer #3: All comments have been addressed

2. Is the manuscript technically sound, and do the data support the conclusions?

Reviewer #2: Yes

Reviewer #3: Yes

3. Has the statistical analysis been performed appropriately and rigorously? 

Reviewer #2: I Don't Know

Reviewer #3: Yes

4. Have the authors made all data underlying the findings in their manuscript fully available?

Reviewer #2: Yes

Reviewer #3: Yes

5. Is the manuscript presented in an intelligible fashion and written in standard English?

Reviewer #2: Yes

Reviewer #3: Yes

6. Review Comments to the Author

Reviewer #2: The authors have made appropriate adjustments regarding the reviewer inputs. Minor details still need to be included. However, adding them depends on the journal's publication topic.

- The WHO reference was still based on the revised 2016, while the new edition of WHO 2024 is already available.

- Thank you for pointing out the RTU abbreviation, but the incubation details have not been included. Different heat and incubation duration options could be unique for various antibodies.

- Thank you for the additional references for comparing the IHC technique with the others. However, the discussion has yet to include this study's limitation regarding IHC use, which might not always depict the gene rearrangement in the tumor cell.

Reviewer #3: I think that the paper does not contain relevant novel data that are woth publishing in a journal like PLOS. The clinical cohort is small, heterogeneous (mixture of all IPI groups), not from a registry or a trial but a single center collection. There is not even a correlation with the "golstandard" FISH, as far as I see. The issue of BCL2/MYC expression has been studied in much larger and more homogeneous cohorts and I simply do not understand what is added to the literature with this manuscript.

7. PLOS authors have the option to publish the peer review history of their article (what does this mean?). If published, this will include your full peer review and any attached files.

Reviewer #2: No

Reviewer #3: No

---

## [Author Response · Author response to Decision Letter 1]

14 Jun 2024

Dear Editor:

We would like to express our gratitude to the reviewers for taking the time to read and provide valuable feedback like this. Below is our reply to your comment.

The study group is small and diverse, collected from a single center without a correlation to the "gold standard" FISH. I think a broad revision could overcome this problematic errors. Please consider adding the nation of origin to the title and emphasizing the limited scope of the study in the introduction. Referring to the most recent version is necessary to indicate the study's restrictions at the time of publishing.

- We considered adding the nation of origin to the title “Immunohistochemistry-based investigation of MYC, BCL2, and Ki-67 protein expression and their clinical impact in diffuse large B-cell lymphoma in Upper Northern Thailand”

- We Added a new paragraph regarding WHO-HAEM5 and the limited scope of the study in the introduction section. (page 5, lines 88-102)

Reviewers:

Reviewer #2: The authors have made appropriate adjustments regarding the reviewer inputs. Minor details still need to be included. However, adding them depends on the journal's publication topic. 

The WHO reference was still based on the revised 2016, while the new edition of WHO 2024 is already available.

- Thank you for raising this issue. We have already added the paragraph that mentions the WHO-HAEM5 in the introduction section. (page 5, lines 88-102)

Thank you for pointing out the RTU abbreviation, but the incubation details have not been included. Different heat and incubation duration options could be unique for various antibodies.

- The incubation time was 16 minutes for MYC and 32 minutes for BCL2 and Ki-67. (page 8, lines 156-157)

Thank you for the additional references for comparing the IHC technique with the others. However, the discussion has yet to include this study's limitation regarding IHC use, which might not always depict the gene rearrangement in the tumor cell.

- We have added the limitation regarding IHC use in the discussion section (page 23, lines 400-401)

Reviewer #3: I think that the paper does not contain relevant novel data that are woth publishing in a journal like PLOS. The clinical cohort is small, heterogeneous (mixture of all IPI groups), not from a registry or a trial but a single center collection. There is not even a correlation with the "golstandard" FISH, as far as I see. The issue of BCL2/MYC expression has been studied in much larger and more homogeneous cohorts and I simply do not understand what is added to the literature with this manuscript.

- Thank you. We understand the issues you have highlighted. This study could not utilize the gold standard FISH technique due to limited resources and the heterogeneous cohort. However, our findings are significant for providing prognostic information. Double expressor can help classify disease prognosis in resource-limited settings. Although FISH is the gold standard according to WHO-HAEM5, the latest Thai Lymphoma Guideline 2022 stated that in cases where the FISH technique cannot be used, IHC detects MYC and BCL2 expression, referred to as double expression, which may correlate with a poorer prognosis than those without or with single expression. Our study demonstrates the value of the IHC technique in distinguishing different expression groups. This allows us to classify patients based on survival outcomes. Additionally, this finding can serve as basic information for Thai patients. We also recognize our study's limitations, which we have already discussed in the discussion section.

---

## [Decision Letter · Decision Letter 2]

3 Jul 2024

Immunohistochemistry-based investigation of MYC, BCL2, and Ki-67 protein expression and their clinical impact in diffuse large B-cell lymphoma in Upper Northern Thailand

PONE-D-23-39686R2

Dear Dr. Daroontum,

We’re pleased to inform you that your manuscript has been judged scientifically suitable for publication and will be formally accepted for publication once it meets all outstanding technical requirements.

Kind regards,

Mehmet Baysal

Academic Editor

PLOS ONE

Additional Editor Comments (optional):

Reviewers' comments:

Reviewer's Responses to Questions

**Comments to the Author**

1. If the authors have adequately addressed your comments raised in a previous round of review and you feel that this manuscript is now acceptable for publication, you may indicate that here to bypass the “Comments to the Author” section, enter your conflict of interest statement in the “Confidential to Editor” section, and submit your "Accept" recommendation.

Reviewer #2: All comments have been addressed

2. Is the manuscript technically sound, and do the data support the conclusions?

Reviewer #2: Yes

3. Has the statistical analysis been performed appropriately and rigorously? 

Reviewer #2: I Don't Know

4. Have the authors made all data underlying the findings in their manuscript fully available?

Reviewer #2: Yes

5. Is the manuscript presented in an intelligible fashion and written in standard English?

Reviewer #2: Yes

6. Review Comments to the Author

Reviewer #2: The authors have made appropriate adjustments regarding the suggestions. Thank you for the efforts.

7. PLOS authors have the option to publish the peer review history of their article (what does this mean?). If published, this will include your full peer review and any attached files.

Reviewer #2: No

---

## [Editor Report · Acceptance letter]

11 Jul 2024

PONE-D-23-39686R2 

PLOS ONE

Dear Dr. Daroontum, 

I'm pleased to inform you that your manuscript has been deemed suitable for publication in PLOS ONE. Congratulations! Your manuscript is now being handed over to our production team.

Kind regards, 

on behalf of

Dr. Mehmet Baysal 

Academic Editor

PLOS ONE